# TRPML1 gating modulation by allosteric mutations and lipids

**Ninghai Gan[1,2], Yan Han[2], Weizhong Zeng[1,2], Youxing Jiang[1,2]***

[1]Howard Hughes Medical Institute and Department of Physiology, University of Texas Southwestern Medical Center, Dallas, United States; [2]Department of Biophysics, University of Texas Southwestern Medical Center, Dallas, United States

***For correspondence:**
youxing.jiang@utsouthwestern.edu

**Competing interest:** The authors declare that no competing interests exist.

**Abstract** Transient Receptor Potential Mucolipin 1 (TRPML1) is a lysosomal cation channel whose loss-of-function mutations directly cause the lysosomal storage disorder mucolipidosis type IV (MLIV). TRPML1 can be allosterically regulated by various ligands including natural lipids and small synthetic molecules and the channel undergoes a global movement propagated from ligand-induced local conformational changes upon activation. In this study, we identified a functionally critical residue, Tyr404, at the C-terminus of the S4 helix, whose mutations to tryptophan and alanine yield gain- and loss-of-function channels, respectively. These allosteric mutations mimic the ligand activation or inhibition of the TRPML1 channel without interfering with ligand binding and both mutant channels are susceptible to agonist or antagonist modulation, making them better targets for screening potent TRPML1 activators and inhibitors. We also determined the high-resolution structure of TRPML1 in complex with the $PI(4,5)P_2$ inhibitor, revealing the structural basis underlying this lipid inhibition. In addition, an endogenous phospholipid likely from sphingomyelin is identified in the $PI(4,5)P_2$-bound TRPML1 structure at the same hotspot for agonists and antagonists, providing a plausible structural explanation for the inhibitory effect of sphingomyelin on agonist activation.

## eLife Assessment

Transient receptor potential mucolipin 1 (TRPML1) functions as a lysosomal ion channel whose variants are associated with lysosomal storage disorder mucolipidosis type IV. This **important** report describes local and global structural changes driven by binding of regulatory phospholipids and by mutations that allosterically cause gain or loss of channel function. Most of the claims related to the allosteric regulation of TRPML1 are **convincingly** supported by two new cryo-EM structures which are evaluated within the context of previously reported TRPML1 structures, and a proposed allosteric gating mechanism is partially supported by functional electrophysiology results.

## Introduction

Transient Receptor Potential Mucolipin 1 (TRPML1) is a $Ca^{2+}$-permeable, non-selective, lysosomal cation channel ubiquitously expressed in mammalian cells (*Dong et al., 2008*; *LaPlante et al., 2002*; *Sun et al., 2000*). TRPML1 plays critical roles in many important cellular activities including lipid accumulation (*Shen et al., 2012*), signaling transduction (*Kilpatrick et al., 2016*), lysosome trafficking (*Venkatachalam et al., 2015*), and autophagy (*Scotto Rosato et al., 2019*). The loss-of-function mutations in TRPML1 directly cause the lysosomal storage disorder mucolipidosis type IV (MLIV), a neurodegenerative disease characterized by abnormal neurodevelopment, retinal degeneration, and iron-deficiency anemia (*Bargal et al., 2000*; *Bassi et al., 2000*; *Gan and Jiang, 2022*; *Nilius et al., 2007*). Because of its physiological importance and direct disease association, TRPML1 has been extensively studied and is a potential target for drug development.

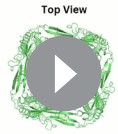

**Video 1.** Conformational changes between open and closed TRPML1.
https://elifesciences.org/articles/100987/figures#video1

TRPML1 can be regulated by various ligands including both natural lipids and small synthetic molecules. The channel can be activated by the lysosome-specific phosphatidylinositol 3,5-bisphosphate ($PI(3,5)P_2$) (*Dong et al., 2010*), but inhibited by the plasma membrane-enriched $PI(4,5)P_2$ (*Zhang et al., 2012*). Given its pharmacological importance, many synthetic agonists and antagonists have been developed for TRPML1 activation and inhibition (*Chen et al., 2014*; *Grimm et al., 2010*; *Samie et al., 2013*; *Shen et al., 2012*). Interestingly, the mTOR (Mammalian target of rapamycin) inhibitor rapamycin and its derivatives can also synergistically activate TRPML1 with $PI(3,5)P_2$ (*Gan et al., 2022*; *Zhang et al., 2019*). Recent studies also suggest that sphingomyelin, a major membrane component, can also modulate the TRPML1 activation (*Prat Castro et al., 2022*; *Shen et al., 2012*).

Several TRPML1 channel structures in both open and closed conformations with various ligands have been determined (*Chen et al., 2017*; *Fine et al., 2018*; *Gan et al., 2022*; *Schmiege et al., 2017*; *Schmiege et al., 2021*), revealing some unique features of the TRPML1 channel. Firstly, all ligand-binding sites in the structures converge to two hot spots: The N-terminal poly-basic pocket for $PIP_2$ and the inter-subunit interface in the middle of the membrane between S5 and S6 for agonists, antagonists, and rapamycin (*Fine et al., 2018*; *Gan et al., 2022*; *Schmiege et al., 2017*; *Schmiege et al., 2021*; *Figure 1a*). Secondly, all open TRPML1 structures are almost identical regardless of the activation stimuli. Thirdly, structural comparison between the open and closed conformation illustrates that TRPML1 gating is not merely a local conformational change but involves the global movement of almost the entire channel mediated by tight inter- and intra-subunit packing within the channel tetramer (*Video 1*). Finally, the necessity of global movement for channel activation underlies the allosteric regulation of TRPML1 by two distantly bound ligands - that is, the ligand-induced local conformational change at one site can propagate to the other site and thereby affect the binding of the other ligand (*Gan et al., 2022*). The high allostery of TRPML1 gating would allow us to design allosteric mutations that are remote from the channel pore but can still stabilize the channel in an open or closed state, mimicking the ligand activation or inhibition of the channel. To this end, we identified Tyr404 on the S4 helix as an allosteric site whose mutation can promote or inhibit TRPML1 gating. Furthermore, we also determined a high-resolution structure of $PI(4,5)P_2$-inhibited TRPML1 and demonstrated that in addition to competing against $PI(3,5)P_2$ activator for the same site, $PI(4,5)P_2$ also allosteric inhibits small molecule agonist by stabilize the channel in the closed conformation. Furthermore, the high-resolution $PI(4,5)P_2$-bound TRPML1 structure also revealed a bound phospholipid likely from sphingomyelin at the agonist/antagonist site, providing a plausible explanation for sphingomyelin inhibition of TRPML1.

## Results

### Allosteric mutations at Tyr404 recapitulate TRPML1 gating

Our previous study on the allosteric activation of TRPML1 by $PI(3,5)P_2$ and rapamycin demonstrated that ligand-induced local conformational changes can propagate to distal parts of the channel through tight inter- and intra-subunit packing within the channel tetramer, allowing the channel to integrate the stimuli from these two distantly bound ligands. The $PI(3,5)P_2$ and rapamycin-induced local conformational changes converge to the same driving force on S4 helix, resulting in a slight bend of the C-terminal half of the S4 that facilitates the channel opening (*Gan et al., 2022*; *Figure 1b*). A key interaction coupled to the S4 bending movement is the insertion of Tyr404 side chain into a pocket surrounded by S1, S3, and S4 helices where its aromatic ring is sandwiched between the side chains of Leu66 and Arg403. We hypothesized that mutations at Tyr404 that stabilize its sidechain in the pocket would facilitate channel activation; conversely, mutations that destabilize its sidechain in the pocket would negatively modulate the channel activation. To test this, we replaced Tyr404 with tryptophan

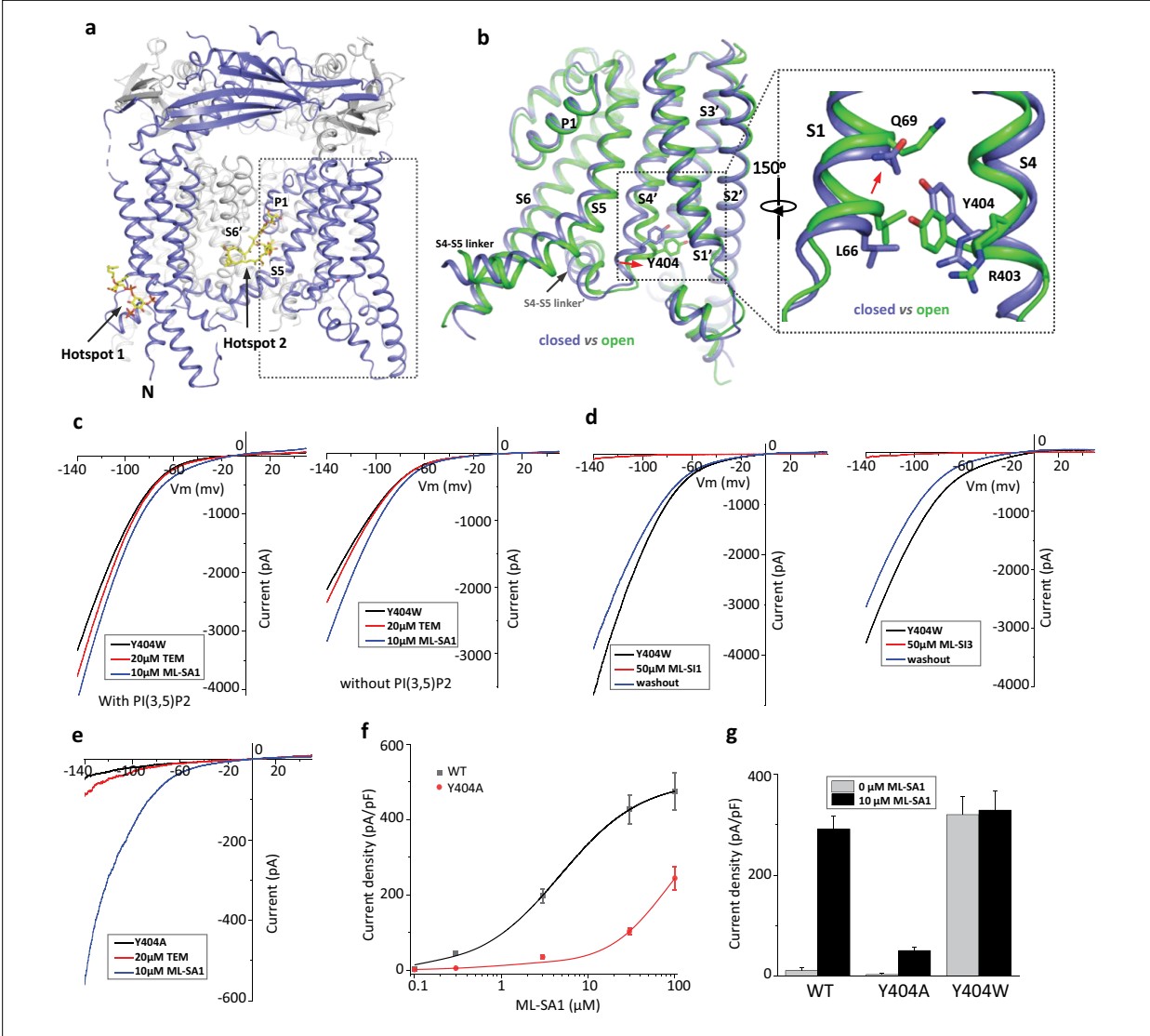

**Figure 1.** Design and characterization of allosteric mutations at Tyr404 that recapitulate TRPML1 gating. (**a**) The structure of $PI(3,5)P_2$/Temsirolimus-activated TRPML1 (PDB code:7SQ9) illustrating the two hot spots for ligand binding. Temsirolimus (Tem) is a rapamycin analog. (**b**) Ligand-induced conformational change and the zoomed-in view of the Y404 movement. Only the boxed region in (**a**) is shown in the structural comparison between the open (green) and closed (blue) structures. Red arrows mark the bending of S4 and upward movement of S1. (**c**) Sample traces of Y404W gain-of-function mutant recorded using patch clamp in whole-cell configuration with (left) or without (right) 100 μM $PI(3,5)P_2$ in the pipette (cytosolic). Tem or agonist ML-SA1 was introduced in the bath solution (extracellular/luminal). (**d**) Sample traces of Y404W inhibition by antagonists ML-SI1 (left) and ML-SI3 (right) recorded using patch clamp in whole-cell configuration. The antagonists were introduced in the bath solution (extracellular/luminal). (**e**) Sample traces of Y404A loss-of-function mutant with 100 μM $PI(3,5)P_2$ in the pipette (cytosolic). Tem or ML-SA1was introduced in the bath solution (extracellular/luminal). (**f**) ML-SA1 activation of TRPML1(WT) and Y404A mutant measured at –140 mV. Data for WT is least square fits to the Hill equation with $EC_{50}$=4.8 ± 0.7 μM, n=0.93 ± 0.10. Data points are mean ± SEM (n=5 independent experiments). (**g**) Current density of wild-type and mutant TRPML1 at –140 mV with and without 10 μM ML-SA1. Data points are mean ± SEM (n=5 independent experiments).

The online version of this article includes the following figure supplement(s) for figure 1:

**Figure supplement 1.** Time course plots of current amplitudes of Y404 mutations recorded at –140 mV with symmetrical pH of 7.4.

and alanine, respectively, and measured the effect of these mutations on channel activity. As illustrated in the electrophysiological recordings using whole-cell patches, the Y404W mutant elicits large inward-rectifying currents without any ligands, indicating that Y404W is a gain-of-function (GOF) mutant (*Figure 1c* and *Figure 1—figure supplement 1a*). Adding extra activation ligands such as $PI(3,5)P_2$, rapamycin, or small molecule agonist ML-SA1 only marginally increases the currents. The Y404W GOF mutant mimics a ligand-activated channel, yet its mutation site is remote from the pore

domain and the channel can still be allosterically inhibited by small molecule antagonists (ML-SI1 and ML-SI3; *Figure 1d* and *Figure 1—figure supplement 1b*). This is distinct from other gain-of-function mutants in which proline substitutions on the S5 helix lock the pore in an open state and the channels are no longer susceptible to antagonist inhibition (*Dong et al., 2009*; *Grimm et al., 2007*; *Kim et al., 2007*; *Nagata et al., 2008*; *Xu et al., 2007*).

Y404A, on the other hand, represents a loss-of-function mutant and elicits much lower currents even in the presence of potent agonist ML-SA1 (*Figure 1e* and *Figure 1—figure supplement 1c*). While ML-SA1 can potently activate the wild-type TRPML1 channel, the Y404A mutation mimics PI(4,5)P$_2$ inhibition and allosterically inhibits ML-SA1 binding, significantly decreasing the efficacy of ML-SA1 activation (*Figure 1f & g*).

## Structure of GOF Y404W mutant

To reveal the structural basis underlying the channel activation of the Y404W mutant, we determined its structure in the absence of any ligands to 2.86 Å resolution (*Figure 2—figure supplements 1–2*, *Supplementary file 1* and Materials and methods). As expected, the Y404W mutant adopts an open conformation with a structure almost identical to other ligand-activated open TRPML1, consistent with its GOF property (*Figure 2a* and *Figure 2—figure supplement 2b*). Like Tyr404 in the wide-type open TRPML1, the side chain of W404 in the mutant is inserted into the pocket surrounded by S1, S3, and S4 helices and sandwiched between Leu66 and Arg403 (*Figure 2b*). However, the larger indole ring of Trp404 provides a better spatial fitting into the pocket than the phenol ring of Tyr404 and several surrounding residues (Lys65, Gln69, and Leu358) provide extra van der Waals contacts to the Trp404 side chain. Thus, by enhancing the stability of the aromatic side chain inside the pocket, Y404W mutation facilitates the bending of S4 which in turn propagates to the pore through the S4-S5 linker and activates the channel (*Gan et al., 2022*). The Y404W mutant structure demonstrates that the sidechain packing in the pocket is essential for stabilizing the open channel and the lack of such packing capacity in the Y404A mutant with a small side-chain likely destabilizes the open conformation, yielding a loss-of-function channel. It is worth noting that Arg403 plays two essential roles in TRPML1 gating: its side chain is part of the pocket that stabilizes Tyr404 in the open state; its guanidinium group forms a salt bridge with the C3 phosphate group of PI(3,5)P$_2$ upon ligand activation (*Gan et al., 2022*). As expected, Arg403 is highly conserved in the TRPML channel family, and its R403C variant identified in an MLIV patient is a loss-of-function mutant (*Chen et al., 2014*).

## Structure of TRPML1 in PI(4,5)P$_2$-bound closed state

While PI(4,5)P$_2$ inhibits PI(3,5)P$_2$ activation of TRPML1 by directly competing for the same binding site, it also allosterically inhibits the agonist-activated channel (*Chen et al., 2017*), suggesting that PI(4,5)P$_2$ binding stabilizes the TRPML1 channel in a closed conformation. A previous low-resolution structure of TRPML1 in complex with PI(4,5)P$_2$ revealed the approximate location of PI(4,5)P$_2$ binding but failed to explain how its binding stabilizes the channel in the closed state and allosterically inhibits the agonist-activated channel (*Fine et al., 2018*). To address this, we determined the structure of PI(4,5)P$_2$-bound TRPML1 at 2.46 Å (*Figure 3a*, *Figure 2—figure supplement 2*, *Figure 3—figure supplements 1–2*, *Supplementary file 1* and Materials and methods). The density from the IP3 head group of PI(4,5)P$_2$, especially the phosphate groups on C4 and C5 of the inositol, can be clearly defined in the EM map (*Figure 3a–c*). The phosphatidyl group, however, is flexible and could not be resolved in the structure. While PI(4,5)P$_2$ binding overlaps with that of PI(3,5)P$_2$, their IP3 head group positions are quite different (*Figure 3b–e*). In the PI(3,5)P$_2$-bound structure (*Figure 3d*), the head group protrudes deep into the N-terminal PIP$_2$-binding pocket enclosed by two short clamp-shaped helices of H1 and H2, and the cytosolic ends of S1 and S2 helices, allowing its C3 phosphate to engage in direct interactions with Arg403 and Tyr355 to facilitate channel activation (*Gan et al., 2022*). These C3 phosphate-mediated interactions are absent in PI(4,5)P$_2$-bound structure. Instead, the head group of PI(4,5)P$_2$ is trapped at the entrance of the pocket and forms a bridge between S1 and S2 with its phosphate groups stabilized by positively charged residues from H2, S1, and S2 (*Figure 3b and c*). A major conformational change between the open and closed states is an upward movement of the S1 helix, a prerequisite for Tyr404 insertion between Leu66 and Arg403 and the subsequent bending of S4 (*Figure 3e*). Therefore, the PI(4,5)P$_2$-mediated bridging interaction between S1 and S2 would

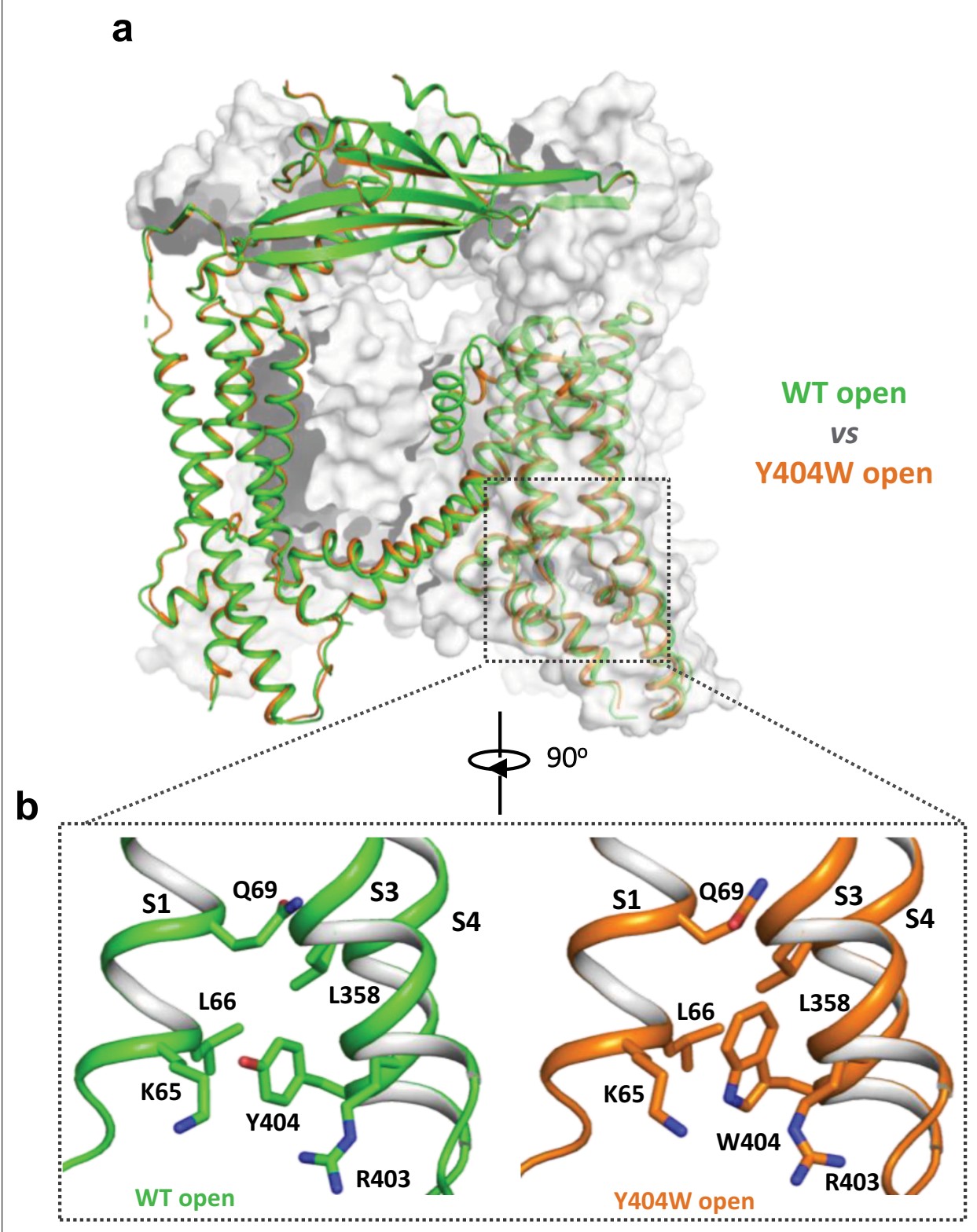

**Figure 2.** Y404W mutant adopts an open conformation in the absence of ligands. (**a**) Structural comparison between PI(3,5)P$_2$/Tem-bound open structure (green) and the Y404W mutant structure (orange). Only the front subunit and the neighboring S1-S4 regions are highlighted in color for clarity. (**b**) Zoomed-in views of the regions surrounding Y404 (WT, green) and W404 (mutant, orange).

The online version of this article includes the following figure supplement(s) for figure 2:

*Figure 2 continued on next page*

hinder the S1 movement and stabilize the channel in the closed conformation, exerting allosteric inhibition on agonist activation.

## Endogenous sphingomyelin lipid at the agonist- and antagonist-binding site

The high-resolution structure of $PI(4,5)P_2$-bound closed TRPML1 also reveals a well-defined density from an endogenous lipid molecule at the inter-subunit interface between S5 and S6 (*Figure 4a*). The lipid contains a choline head group and is likely a phosphatidylcholine (PC) or sphingomyelin (SM), the two main choline-containing phospholipid components of the outer leaflet of the plasma membrane. The tail from one of the lipid alkyl chains penetrates deep into an inter-subunit pocket in the middle of the membrane, overlapping with the hotspot for both channel agonist and antagonist (*Figure 4—figure supplement 1*). This alkyl chain has to be displaced upon agonist or antagonist binding, suggesting that the lipid occupation would compete against agonist or antagonist binding. We suspect this bound lipid is sphingomyelin which is also enriched in the endocytic recycling compartment and has been shown to inhibit TRPML1 activity (*Prat Castro et al., 2022*; *Schuchman, 2010*; *Shen et al., 2012*; *Slotte, 2013*). Key evidence to support SM inhibition is that its enrichment can reduce the agonist (i.e. SF-51 and ML-SA1) activation of TRPML1 (*Shen et al., 2012*). Indeed, we did observe the reduction of SF-51-activated TRPML1 current upon SM enrichment (*Figure 4b* and *Figure 4—figure supplement 2a*). However, based on our structure, we hypothesize that the role of sphingomyelin is to stabilize rather than directly inhibit the channel; the SM inhibition upon enrichment is an indirect effect attributable to its competition against agonist binding that reduces the apparent efficacy of agonist activation. This hypothesis would imply that SM can also function as an indirect activator by competing against antagonists and reducing their effectiveness in channel inhibition. The gain-of-function Y404W mutant, which is still susceptible to antagonist inhibition, provides a good system to test that. As shown in *Figure 4c* and *Figure 4—figure supplement 2b*, SM shows no obvious inhibition to the mutant channel activity, whereas antagonist ML-SI1 markedly reduces the mutant channel current; upon SM enrichment, ML-SI1 inhibition is mitigated resulting in a recovery of the channel current. This observation confirms the competitive binding of SM at the hot spot for both agonists and antagonists.

## Discussion

In this study, we designed and analyzed the allosteric mutations at Tyr404 that recapitulate the gating of TRPML1. Replacing this tyrosine with tryptophan or alanine stabilizes or destabilizes the channel in the open state, yielding a gain- or loss-of-function mutant. The structure of the Y404W mutant adopts the same open structure as ligand-activated TRPML1, once again highlighting the global conformational change for TRPML1 channel activation. As Tyr404 is distant from the hot spots for ligand binding, the two gain- and loss-of-function mutants can still be allosterically modulated by antagonists and agonists. Thus, these allosteric mutants can mimic ligand-activated or inhibited TRPML1 without interfering with ligand binding, making them better targets for screening potent small molecule TRPML1 inhibitors and activators. We also investigated the structural basis of $PI(4,5)P_2$ inhibition of TRPML1 by determining the $PI(4,5)P_2$-bound structure, revealing a different binding mode by its head group at the N-terminal polybasic site than that of $PI(3,5)P_2$. The head group of $PI(4,5)P_2$ mediates a bridging interaction between S1 and S2 and stabilizes TRPML1 in a closed conformation. In the high-resolution $PI(4,5)P_2$-bound TRPML1 structure, we also visualize clear density from a choline-containing phospholipid at the same site for agonists or antagonists. In light of its high membrane abundance and competing effect on agonist activation and antagonist inhibition, this bound lipid is likely from sphingomyelin.

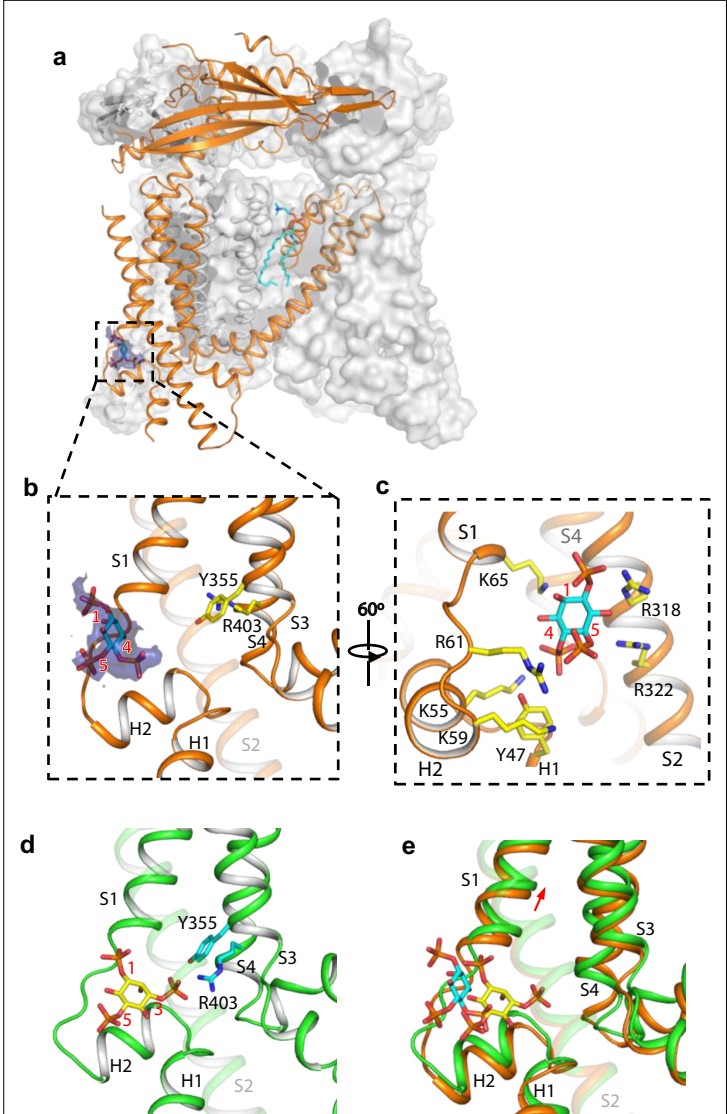

**Figure 3.** Structure of TRPML1 in complex with PI(4,5)P$_2$. (**a**) Overall structure of PI(4,5)P$_2$-bound TRPML1 with the front subunit shown in orange cartoon and the rest shown as grey surface representation. Density for PI(4,5)P$_2$ head group is shown in blue surface. (**b**) Zoomed-in view of the PI(4,5)P$_2$-binding pocket with the density of its IP3 head group shown in blue surface. (**c**) Zoomed-in view of the PI(4,5)P$_2$-binding pocket with side chains of IP3-interacting residues shown as yellow sticks. (**d**) Zoomed-in view of the IP3 position in the PI(3,5)P$_2$-bound open TRPML1 structure. The C3 phosphate group directly interacts with Y355 and R403. (**e**) Comparison of the head group positions in PI(3,5)P$_2$-bound open (green) and PI(4,5)P$_2$-bound closed (orange) structures. The inositol rings PI(3,5)P$_2$ and PI(4,5)P$_2$ are colored yellow and cyan, respectively. The red arrow marks the upward movement of S1 from closed to open conformation.

The online version of this article includes the following figure supplement(s) for figure 3:

**Figure supplement 1.** Cryo-EM data processing scheme of the TRPML1 sample prepared in the presence of PI(4,5)P$_2$.

**Figure supplement 2.** Sample density maps of the PI(4,5)P$_2$-bound closed TRPML1 structure contoured at 4 σ.

# Materials and methods

## Key resources table

| Reagent type (species) or resource | Designation | Source or reference | Identifiers | Additional information |
|---|---|---|---|---|
| Strain, strain background (*Escherichia coli*) | TOP10 | Thermo Fisher Scientific | Cat# 18258012 | |
| Strain, strain background (*E. coli*) | DH10bac | Thermo Fisher Scientific | Cat# 10361012 | |
| Cell line (*Spodoptera frugiperda*) | Sf9 cells | Thermo Fisher Scientific | Cat# 11496015; RRID:CVCL_0549 | |
| Cell line (*Homo sapiens*) | FreeStyle 293 F cells | Thermo Fisher Scientific | Cat# R79007; RRID:CVCL_D603 | |
| Transfected construct (*H. sapiens*) | pEZT-BM-mTRPML1-CHis | This paper | N/A | Construct made to express the protein in HEK293F cells |
| Recombinant DNA reagent | pEZT-BM | DOI:10.1016 /j. str.2016.03.004 | Addgene:74099 | |
| Sequence-based reagent | *Mcoln1*_F_primer: cgCTCGAG gccgccaccATGGCC ACCCCGGCGGGC | Integrated DNA Technologies | N/A | |
| Sequence-based reagent | *Mcoln1*_R_primer: at gcggccgcTCAGTTC ACCAGCAGCGA | Integrated DNA Technologies | N/A | |
| Sequence-based reagent | *Mcoln1*_Y404A_F_primer: cttgtggaaaaatgtcaggg cgcgaatgacaccgacccag | Integrated DNA Technologies | N/A | |
| Sequence-based reagent | *Mcoln1*_Y404A_R_primer: ctgggtcggtgtcattcgcg ccctgacatttttccacaag | Integrated DNA Technologies | N/A | |
| Sequence-based reagent | *Mcoln1*_Y404W_F_primer: cttgtggaaaaatgtcag ccagcgaatgacaccgaccc | Integrated DNA Technologies | N/A | |
| Sequence-based reagent | *Mcoln1*_Y404W_R_primer: gggtcggtgtcattcgctg gctgacattttccacaag | Integrated DNA Technologies | N/A | |
| Chemical compound, drug | Sodium Butyrate | Sigma-Aldrich | Cat# 303410 | |
| Chemical compound, drug | n-dodecyl-β-D-maltopyranoside | Anatrace | Cat# D310 | |
| Chemical compound, drug | glyco-diosgenin | Anatrace | Cat# GDN101 | |
| Chemical compound, drug | ML-SA1 | Sigma-Aldrich | Cat# SML0627 | |
| Chemical compound, drug | ML-SI1 | Medchemexpress | Cat# HY-134818 | |
| Chemical compound, drug | PI(4,5)P2 diC8 | Echelon | Cat# P-4508 | |
| Chemical compound, drug | Sphingomyelin | Sigma-Aldrich | Cat# 567706 | |
| Chemical compound, drug | Temsirolimus | Fisher Scientific | Cat# 52-641-0 | |
| Chemical compound, drug | ML-SI3 | Selleckchem | Cat# E0026 | |
| Chemical compound, drug | SF-51 | Chemspace | Cat# CSSS00121681914 | |
| Chemical compound, drug | Thrombin | Sigma-Aldrich | Cat# T4648 | |

*Continued on next page*

*Continued*

| Reagent type (species) or resource | Designation | Source or reference | Identifiers | Additional information |
|---|---|---|---|---|
| Software, algorithm | MotionCor2 | *Zheng et al., 2017* | | https://emcore.ucsf.edu/ucsf-software |
| Software, algorithm | GCTF | *Zhang, 2016; JackZhang-Lab, 2021b* | | https://github.com/JackZhang-Lab/GCTF |
| Software, algorithm | RELION | *Scheres, 2012* | | http://www2.mrc-lmb.cam.ac.uk/relion |
| Software, algorithm | Chimera | *Pettersen et al., 2004* | RRID:SCR_004097 | https://www.cgl.ucsf.edu/chimera |
| Software, algorithm | PyMol | Schrödinger | RRID:SCR_000305 | https://pymol.org/2 |
| Software, algorithm | COOT | *Emsley et al., 2010* | RRID:SCR_014222 | https://www2.mrc-lmb.cam.ac.uk/personal/pemsley/coot |
| Software, algorithm | MolProbity | *Chen et al., 2010* | | http://molprobity.biochem.duke.edu/ |
| Software, algorithm | PHENIX | *Adams et al., 2010* | | https://www.phenix-online.org |
| Other | Superose 6 Increase10/300 GL | GE Healthcare | Cat# 29091596 | Used to perform gel filtration |
| Other | Ni-NTA Agarose | Qiagen | Cat# 30210 | Used to purify His-tagged protein |
| Other | Amicon Ultra-15 Centrifugal Filter Units | Milliporesigma | Cat# UFC9100 | Used to concentrate protein sample |
| Other | Quantifoil R 1.2/1.3 grid Au300 | Quantifoil | Cat# Q37572 | Used to prepare cryoEM samples |
| Commercial assay or kit | Cellfectin | Thermo Fisher Scientific | Cat# 10362100 | |
| Other | Sf-900 II SFM medium | Thermo Fisher Scientific | Cat# 10902088 | Used to culture SF9 cells |
| Other | FreeStyle 293 Expression Medium | Thermo Fisher Scientific | Cat# 12338018 | Used to culture HEK293F cells |
| Chemical compound, drug | Antibiotic Antimycotic Solution | Sigma-Aldrich | Cat# A5955 | |
| Chemical compound, drug | Proteinase K | Thermo Fisher Scientific | Cat# EO0491 | |
| Commercial assay or kit | Lipofectamine 2000 | Thermo Fisher Scientific | Cat# 11668027 | |

## Protein expression and purification

Protein expression and purification were performed as previously described (*Gan et al., 2022*). The *Mus musculus Mcoln1* gene with a C-terminal thrombin cleavage site and a 10×His tag was cloned into a pEZTBM vector (*Morales-Perez et al., 2016*) and heterologously expressed in HEK293F cells using the BacMam system. The baculovirus was produced in Sf9 cells and used to transduce the HEK293F cells at a ratio of 1:40 (virus:HEK293F, v/v) and supplemented with 1 mM sodium butyrate to boost the protein expression. Cells were cultured in suspension at 37 °C for 48 hr and harvested by centrifugation at 3000 × *g*. All purification procedures were carried out at 4 °C unless specified otherwise. The cell pellet was re-suspended in buffer A (20 mM Tris pH 8.0, 150 mM NaCl) supplemented with a pro0tease inhibitor cocktail (containing 1 mg ml−1 each of DNase, pepstatin, leupeptin, and aprotinin and 1 mM PMSF) and homogenized by sonication on ice. Protein was extracted with 1% (w/v) n-dodecyl-β-D-maltopyranoside (DDM; Anatrace) supplemented with 0.2% (w/v) cholesteryl hemisuccinate (CHS; Sigma-Aldrich) by gentle agitation for 2 hr. After extraction, the supernatant was collected after a 1 hr centrifugation at 48,000 × *g* and incubated with Ni-NTA resin and 20 mM imidazole with gentle agitation. After 1 hr, the resin was collected on a disposable gravity column (Bio-Rad), washed with buffer B (buffer A+0.04%glyco-diosgenin [GDN; Anatrace]) with 20 mM imidazole. The washed resin was left on-column in buffer B and digested with thrombin overnight. After digestion, the flow-through was concentrated, and purified by size-exclusion chromatography on a Superose 6 10/300 GL column (GE Heathcare) pre-equilibrated with buffer B. The protein peak was collected and concentrated. For PI(4,5)$P_2$-bound structure, purified protein was incubated with 0.5 mM PI(4,5)$P_2$ on ice for 4 hr. The lipid ligand used in this study is PI(4,5)$P_2$ diC8 (Echelon) HEK293F cells (RRID:CVCL_D603) were purchased

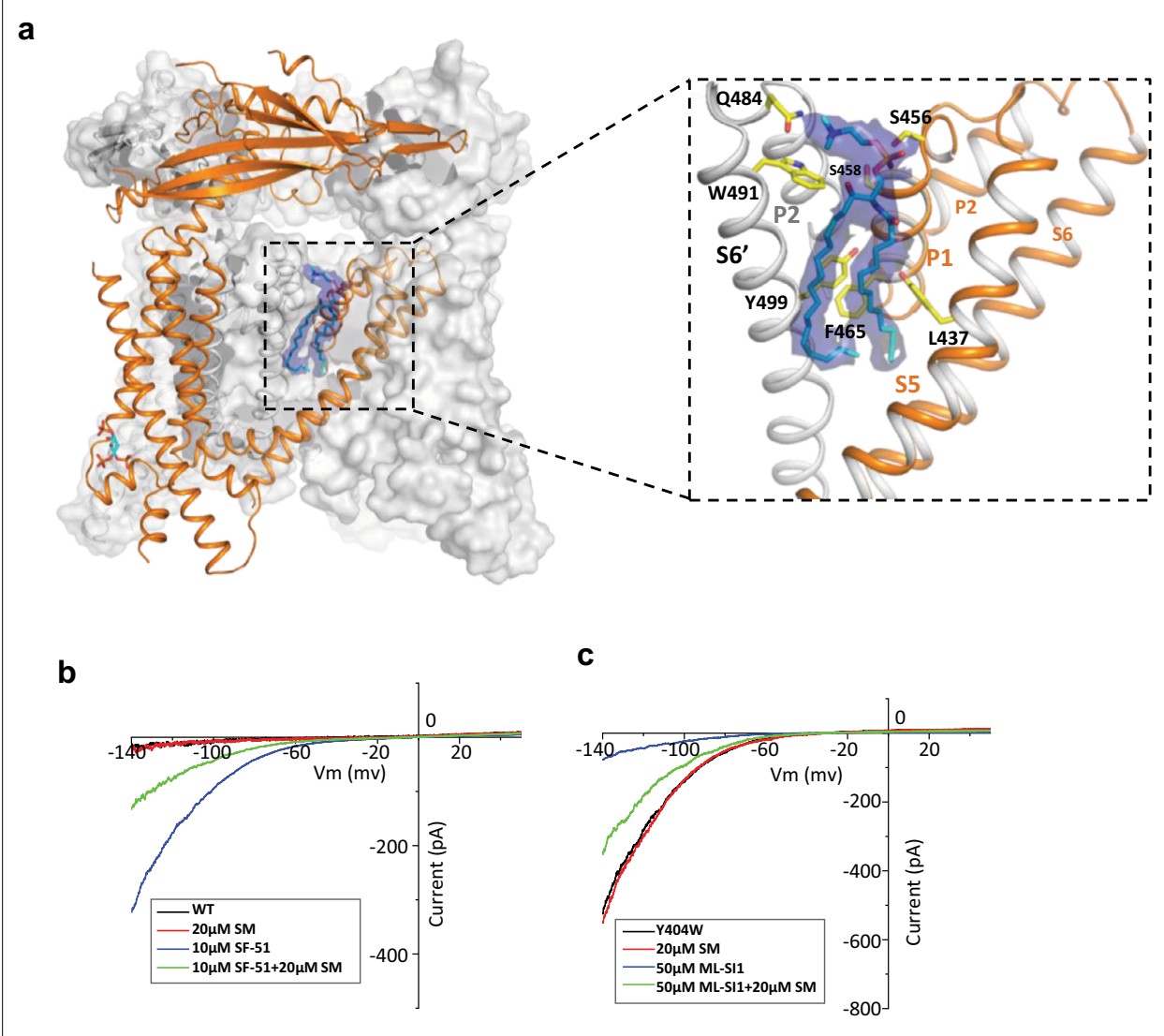

**Figure 4.** Sphingomyelin binding in TRPML1. (**a**) Overall structure of PI(4,5)P$_2$-bound TRPML1 and the zoomed-in view of the lipid-binding site. The lipid density is shown as blue surface and modeled as sphingomyelin (SM). The side chains of lipid-interacting residues are shown as yellow sticks. (**b**) SM inhibition effect on SF-51-activated wild-type TRPML1. (**c**) SM activation effect on ML-SI1-inhibited Y404W mutant. Currents shown in (**b**) and (**c**) were recorded using patch clamp in whole-cell configuration with pH 4.6 in the bath solution as the adverse effect of SM on agonist or antagonist is subtle and is measurable only at low luminal pH.

The online version of this article includes the following figure supplement(s) for figure 4:

**Figure supplement 1.** Sphingomyelin (cyan) binding overlaps with that of agonist ML-SA1 (yellow), rapamycin analog Tem (magenta), or antagonist ML-SI3 (green).

**Figure supplement 2.** Time course plots of sphingomyelin affected TRPML1 current amplitudes.

from and authenticated by Thermo Fisher Scientific. The cell lines tested negative for mycoplasma contamination.

## Electron microscopy data acquisition

Electron microscopy data acquisition followed the protocol previously described (*Gan et al., 2022*). The cryo-EM grids were prepared by applying 3.5 μl protein (3.5 mg/mL) to a glow-discharged Quantifoil R1.2/1.3 200-mesh copper holey carbon grid (Quantifoil, Micro Tools GmbH) and blotted for 3.0 s under 100% humidity at 4 °C before being plunged into liquid ethane using a Mark IV Vitrobot (FEI). For the dataset of Y404W, micrographs were acquired on a Titan Krios microscope (FEI) operated

at 300 kV with a K3 Summit direct electron detector (Gatan), using a slit width of 20 eV on a GIF-Quantum energy filter. Data were collected using CDS (Correlated Double Sampling) mode of the K3 camera with a super resolution pixel size of 0.413 Å. The defocus range was set from −0.9 to −2.2 μm. Each movie was dose-fractionated to 60 frames with a dose rate of 1e-/$Å^2$/frame for a total dose of 60e-/$Å^2$. The total exposure time was between 5 and 6 s. For the PI(4,5)$P_2$-bound dataset, micrographs were acquired on a Titan Krios microscope (FEI) operated at 300 kV with a Falcon4 electron detector (Thermo Fisher), using a slit width of 20 eV on a post-column Selectris X energy filter (Thermo Fisher Scientific). Data was collected using Falcon 4 camera with a pixel size of 0.738 Å. The defocus range was set from −0.9 to −2.2 μm. Each movie was dose-fractionated to 60 frames with a dose rate of 1e-/$Å^2$/frame for a total dose of 60e-/$Å^2$. The total exposure time was between 3.5 and 4 s.

## Image processing

Images were processed as previously described (*Gan et al., 2022*). Movie frames were motion corrected and binned two times and dose-weighted using MotionCor2 (*Zheng et al., 2017*). The CTF parameters of the micrographs were estimated using the GCTF program (*Zhang, 2016*). The rest of the image processing steps were carried out using RELION 3.1 (*Nakane et al., 2020*; *Scheres, 2012*; *Zivanov et al., 2018*). All resolution was reported according to the gold-standard Fourier shell correlation (FSC) using the 0.143 criterion (*Henderson et al., 2012*). Local resolution was estimated using Relion. Aligned micrographs were manually inspected to remove those with ice contamination and bad defocus. Particles were selected using Gautomatch (K. Zhang, MRC LMB, https://github.com/JackZhang-Lab/Gautmatch; *JackZhang-Lab, 2021a*) and extracted using a binning factor of 3. 2D classification was performed in Relion 3.1. Selected particles after 2D classification were subjected to one around 3D classification. The mouse TRPML1 map (EMD-8883 [*Chen et al., 2017*]) low-pass filtered to 30 Å was used as the initial reference. Classes that showed clear features of the TRPML1 channel were combined and subjected to 3D auto-refinement and another round of 3D classification without performing particle alignment using a soft mask around the protein portion of the density. The best resolving classes were then re-extracted with the original pixel size and further refined. Beam tilt, anisotropic magnification, and per-particle CTF estimations and Bayesian polishing were performed in Relion 3.1 to improve the resolution of the final reconstruction.

For the Y404W structure dataset, a total of 4724 movies were collected and 4505 were selected after motion correction and CTF estimation. A total number of 864,698 particles were extracted from the selected micrographs and were subjected to one round of 2D classification, from which 87,846 particles were selected. After the initial 3D classification, 35,460 particles were selected and subjected to a 3D auto-refinement job and further ctf refinements, yielding a map at 2.86 Å overall resolution (*Figure 2—figure supplement 1*).

For the PI(4,5)$P_2$-bound dataset, a total of 8164 movies were collected and 7895 were selected after motion correction and CTF estimation. A total number of 1,065,778 particles were extracted from the selected micrographs and were subjected to one round of 2D classification, from which 555,281 particles were selected. After the initial 3D classification, 359,441 particles were selected and subjected to a 3D auto-refinement job. Next, a soft mask excluding the micelle density was applied and particles were sorted into five classes without performing alignment. From this, one classe with a total number of 60,597 particles were selected and further refined. In the postprocess step, a B-factor of −60 was manually given, yielding a map at 2.46 Å overall resolution (*Figure 3—figure supplement 1*).

## Model building, refinement, and validation

Model building, refinement and validation followed the previously described protocol (*Gan et al., 2022*). The structure of mouse TRPML1 (PDB code: 5WPV) was used as the initial model and was manually adjusted in Coot (*Emsley et al., 2010*) and refined against the map by using the real space refinement module with secondary structure and non-crystallographic symmetry restraints in the Phenix package (*Adams et al., 2010*). The final structure model of Y404W includes residues 40–200, 216–527. The final structure model of the PI(4,5)$P_2$-bound includes residues 39–200, 216–285, 296–527. About 40 residues at the amino terminus and 50 residues at the carboxy terminus are disordered and not modeled. The statistics of the geometries of the models were generated using MolProbity (*Chen*

et al., 2010). All the figures were prepared in PyMol (Schrödinger, LLC.), UCSF Chimera (*Pettersen et al., 2004*). Pore radii were calculated using the HOLE program (*Smart et al., 1996*).

## Electrophysiology

Electrophysiology was carried out following a previously described protocol with minor modifications (*Gan et al., 2022*). For electrophysiological analysis, the two di-leucine motifs ($15_{LL}$ and $577_{LL}$) of mouse TRPML1 responsible for lysosomal targeting were replaced with alanines to facilitate the trafficking of the channel to the plasma membrane (*Grimm et al., 2010*; *Vergarajauregui and Puertollano, 2006*). The N-terminal GFP tagged, plasma membrane-targeting TRPML1 mutant (TRPML1-4A) and derived point mutations were overexpressed in HEK293 cells and the channel activities were directly measured by patching the plasma membrane. In this setting, the extracellular side is equivalent to the luminal side of TRPML1 in endosomes or lysosomes. Forty-eight hr after transfection, cells were dissociated by trypsin treatment and kept in complete serum-containing medium; the cells were re-plated onto 35 mm tissue culture dishes and kept in a tissue culture incubator until recording. Patch clamp in the whole-cell or inside-out configuration was used to measure TRPML1 activity on the HEK plasma membrane. The standard bath solution for whole cell current recording contained (in mM): 145 sodium methanesulfonate, 5 NaCl, 1 $MgCl_2$, 10 HEPES buffered with Tris, pH 7.4; and the pipette solution contained (in mM): 140 caesium methanesulfonate, 5 NaCl, 5 $MgCl_2$, 10 EGTA, 10 HEPES buffered with Tris, pH 7.4. The bath solution for inside-out configuration contained (in mM): 140 potassium methanesulfonate, 5 NaCl, 2 $MgCl_2$, 0.4 $CaCl_2$, 1 EGTA, 10 HEPES buffered with Tris, pH 7.4; and the pipette solution contained (in mM): 145 sodium methanesulfonate, 5 NaCl, 1 $MgCl_2$, 0.5 EGTA, 10 HEPES buffered with Tris, pH 7.4. For whole cell recording of $PI(3,5)P_2$-activated channel, we had to include high concentration of $PI(3,5)P_2$ (100 μM) in the pipette solution (cytosolic side) in order to quickly obtain stable $PI(3,5)P_2$-evoked current, likely because of the slow diffusion of this lipid ligand. $PI(4,5)P_2$ was added in the cytosolic side, Tem, ML-SA1, ML-SI3, ML-SI1, SM were added in the bath solution. SM competition assays with SF-51 and ML-SI1 were conducted under pH 4.6. The patch pipettes were pulled from Borosilicate glassand heat polished to a resistance of 2–5 MΩ (2–3 MΩ for inside-out patch, and 3–5 MΩ for whole-cell current recoding). Data were acquired using an AxoPatch 200B amplifier (Molecular Devices) and a low-pass analogue filter set to 1 kHz. The current signal was sampled at a rate of 20 kHz using a Digidata 1550B digitizer (Molecular Devices) and further analyzed with pClamp 11 software (Molecular Devices). After the patch pipette attached to the cell membrane, the giga seal (>10 GΩ) was formed by gentle suction. The inside-out configuration was formed by pulling the pipette away from the cell, and the pipette tip was exposed to the air for 2 s. The whole-cell configuration was formed by short zap or suction to rupture the patch. The holding potential was set to 0 mV. The whole-cell and inside-out macroscopic current recordings were obtained using voltage pulses ramped from −140 mV to +50 mV over a duration of 800ms. The sample traces for the I–V curves of macroscopic currents shown in each figure were obtained from recordings on the same patch. All data points are mean ±s.e.m. ($n \geq 5$).

## Acknowledgements

Cryo-EM sample grids were prepared at the Structural Biology Laboratory at the University of Texas Southwestern Medical Center which is partially supported by the CPRIT Core Facility Support Award RP170644. Single particle Cryo-EM data were collected at the University of Texas Southwestern Medical Center Cryo-EM Facility that is funded by the CPRIT Core Facility Support Award RP170644 and Pacific Northwest Center for Cryo-EM (PNCC). We thank Omar Davulcu for helping in data collection at PNCC under user proposal 51776. Ninghai Gan is a HHMI fellow of the Jane Coffin Childs Memorial Fund. This work was supported in part by the Howard Hughes Medical Institute and by grants from the National Institute of Health (R35GM140892 to YJ) and the Welch Foundation (Grant I-1578 to YJ).

## Additional information

### Funding

| Funder | Grant reference number | Author |
|---|---|---|
| Howard Hughes Medical Institute | | Youxing Jiang |
| National Institutes of Health | R35GM140892 | Youxing Jiang |
| Welch Foundation | I-1578 | Youxing Jiang |
| Jane Coffin Childs Memorial Fund for Medical Research | | Ninghai Gan |

The funders had no role in study design, data collection and interpretation, or the decision to submit the work for publication.

### Author contributions

Ninghai Gan, Conceptualization, Formal analysis, Validation, Investigation, Methodology, Writing - original draft, Writing - review and editing; Yan Han, Resources, Investigation; Weizhong Zeng, Resources, Formal analysis, Investigation, Methodology; Youxing Jiang, Conceptualization, Resources, Formal analysis, Supervision, Funding acquisition, Methodology, Project administration

### Author ORCIDs

Ninghai Gan ⬤ https://orcid.org/0000-0001-7238-4056
Yan Han ⬤ https://orcid.org/0000-0002-1207-7756
Youxing Jiang ⬤ https://orcid.org/0000-0002-1874-0504

Joint Public Review: https://doi.org/10.7554/eLife.100987.3.sa1
Author response https://doi.org/10.7554/eLife.100987.3.sa2

---

## Additional files

### Supplementary files

• Supplementary file 1. Data collection and refinement statistics.
• MDAR checklist

### Data availability

The cryo-EM density maps of mouse TRPML1 have been deposited in the Electron Microscopy Data Bank (EMDB) under accession numbers 45429 (Y404W), 45432 (PI(4,5)P2-bound). Atomic coordinates have been deposited in the Protein Data Bank (PDB) under accession numbers 9CBZ (Y404W), 9CC2 (PI(4,5)P2-bound).

The following datasets were generated:

| Author(s) | Year | Dataset title | Dataset URL | Database and Identifier |
|---|---|---|---|---|
| Gan N, Jiang Y | 2024 | Cryo-EM structure of mouse TRPML1 channel Y404W at 2.86 Angstrom resolution | https://www.ebi.ac.uk/emdb/EMD-45429 | Electron Microscopy Data Bank, EMD-45429 |
| Gan N, Jiang Y | 2024 | Cryo-EM structure of mouse PI(4,5)P2-bound TRPML1 channel at 2.46 Angstrom resolution | https://www.ebi.ac.uk/emdb/EMD-45432 | Electron Microscopy Data Bank, EMD-45432 |

*Continued on next page*

*Continued*

| Author(s) | Year | Dataset title | Dataset URL | Database and Identifier |
|---|---|---|---|---|
| Gan N, Jiang Y | 2024 | Cryo-EM structure of mouse TRPML1 channel Y404W at 2.86 Angstrom resolution | https://www.rcsb.org/structure/9CBZ | RCSB Protein Data Bank, 9CBZ |
| Gan N, Jiang Y | 2024 | Cryo-EM structure of mouse PI(4,5)P2-bound TRPML1 channel at 2.46 Angstrom resolution | https://www.rcsb.org/structure/9CC2 | RCSB Protein Data Bank, 9CC2 |

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
