## [Editor Report · eLife Assessment]

Transient receptor potential mucolipin 1 (TRPML1) functions as a lysosomal ion channel whose variants are associated with lysosomal storage disorder mucolipidosis type IV. This **important** report describes local and global structural changes driven by binding of regulatory phospholipids and by mutations that allosterically cause gain or loss of channel function. Most of the claims related to the allosteric regulation of TRPML1 are **convincingly** supported by two new cryo-EM structures which are evaluated within the context of previously reported TRPML1 structures, and a proposed allosteric gating mechanism is partially supported by functional electrophysiology results.

---

## [Referee Report · Joint Public Review]

TRPML1 functions as a lysosomal ion channel whose variants are associated with lysosomal storage disorder mucolipidosis type IV. Understanding the structure and function of sites involved in the allosteric control TRPML1 may provide new molecular moieties to target with prototypic drugs.

Gan et al provide the first high resolution cryo-EM structure of a mutant (Y404W) TRPML1 channel in the open state without any activating ligands. This new structure demonstrates how a mutation at a site some distance away from the pore can influence channel gating. The authors provide compelling electrophysiology evidence which supports the proposed Y404W gain of function effect.

The authors propose an allosteric mechanism whereby the engineered W404 sidechain provides extra van der Waals contacts within a pocket surrounded by helices of the voltage sensor-like domain (VSLD) and causes S4 bending which in turn opens the pore through the S4-S5 linker. Conversely, the authors functionally demonstrate that an alanine mutation at this site causes a loss of function. Although the authors do not provide a structure of the Y404A mutant, they propose that the alanine substitution disrupts the sidechain packing and likely destabilizes the open conformation.

TRPML1 channels are regulated by PIP2 species in the cell. In the lysosomal membrane, PI(3,5)P2 activates the channel, whereas in the plasma membrane PI(4,5)P2 inhibits it. Towards understanding its lipid regulation, the authors solve a cryo-EM structure of TRPML1 bound to PI(4,5)P2 in the closed state and provide functional evidence that PI(4,5)P2 occupancy inhibits TRPML1 currents.

Within this same structure, the authors observe a density which may be attributed to sphingomyelin (or possibly phosphocholine). Using electrophysiology on WT and Y404W channels, the authors report an antagonist effect of sphingomyelin on TRPML1 currents.

Taken together, the study provides convincing evidence for a gating (opening/closing) mechanism of the TRPML1 pore which can be allosterically regulated by altered side-chain packing and by lipid interactions within the VSLD.

---

## [Author Response]

The following is the authors’ response to the original reviews.

**Reviewer #1 (Public Review):**
In their manuscript, Gan and colleagues identified a functional critical residue, Tyr404, which when mutated to W or A results in GOF and LOF of TRPML1 activity, respectively. In addition, the authors provide a high-resolution structure of TRPML1 with PI(4,5)P2 inhibitor. This high-resolution structure also revealed a bound phospholipid likely sphingomyelin at the agonist/antagonist site, providing a plausible explanation for sphingomyelin inhibition of TRPML1.This is an interesting study, revealing valuable additional information on TRPML1 gating mechanisms including effects on endogenous phospholipids on channel activity. The provided data are convincing. Some major open questions remain. The work will be of interest to a wide audience including industry researchers occupied with TRPML1 exploration as a drug target.

We appreciate reviewer #1’s positive comments and the specific points raised by this reviewer are addressed in our response to Recommendations For The Authors

**Reviewer #2 (Public Review):**
The transient receptor potential mucolipin 1 (TRPML1) functions as a lysosomal organelle ion channel whose variants are associated with lysosomal storage disorder mucolipidosis type IV. Understanding sites that allosterically control the TRPML1 channel function may provide new molecular moieties to target with prototypic drugs.Gan et al provide the first high-resolution cryo-EM structures of the TRPML1 channel (Y404W) in the open state without any activating ligands. This new structure demonstrates how a mutation at a site some distance away from the pore can influence the channel's conducting state. However, the authors do not provide a structural analysis of the Y404W pore which would validate their open-state claims. Nonetheless, Gan et al provide compelling electrophysiology evidence which supports the proposed Y404W gain of function effect. The authors propose an allosteric mechanism with the following molecular details- the Y404 to W sidechain substitution provides extra van der Waals contacts within the pocket surrounded by helices of the VSD-like domain and causes S4 bending which in turn opens to the pore through the S4-S5 linker. Conversely, the author functionally demonstrates that an alanine mutation at this site causes a loss of function. Although the authors do not provide a structure of the Y404A mutation, they propose that the alanine substitution disrupts the sidechain packing and likely destabilizes the open conformation. TRPM1 channels are regulated by PIP2 species, which is related to their cell function. In the membrane of lysosomes, PI(3,5)P2 activates the channel, whereas PI(4,5)P2 found in the plasma membrane has inhibitory effects. To understand its lipid regulation, the authors solved a cryo-EM structure of TRPM1 bound to PI(4,5)P2 in its presumed closed state. Again, while the provided functional evidence suggests that PI(4,5)P2 occupancy inhibits TRPML1 current, the authors do not provide analysis of the pore which would support their closed state assertion. Within this same structure, the authors observe a density that may be attributed to sphingomyelin (or possibly phosphocholine). Using electrophysiology of WT and the Y404W channels, the authors report sphingomyelins antagonist effect on TRPML1 currents under low luminal (external) pH. Taken together, the results described in Gan et al provide compelling evidence for a gating (open, closed) mechanism of the TRPML1 pore which can be allosterically regulated by altered packing and lipid interactions within the VSDL.

We appreciate reviewer #2’s positive comments and constructive suggestions. We functionally demonstrated that the Y404A mutant is more stable in the closed state. We did not pursue the structure of this mutant as we expect its structure will be the same as the apo closed TRPML1. To verify the open conformation of the Y404W mutant and the closed conformation of PI(4,5)P2 –bound TRPML1, we analyzed the pore radii of our structures in the revision as suggested by the reviewer and compared them with open and closed pores from previously determined TRPML1. Some specific points raised by this reviewer are addressed in our response to Recommendations For The Authors

**Reviewer #1 (Recommendations For The Authors):**
(1) Mutations in TRPML1 cause Mucolipidosis type IV. One patient mutation reported earlier (Chen et al., 2014 https://pubmed.ncbi.nlm.nih.gov/25119295/) to be a LOF mutation is R403C. This mutation resides just next to the here-identified Y404 position which can be converted in either LOF or GOF. Another patient mutation, F408del (also reported previously: (Chen et al., 2014 https://pubmed.ncbi.nlm.nih.gov/25119295/)) results in a mild activity reduction, in particular of the PI(3,5)P2 effect. Can the authors please discuss their findings in the context of the reported literature on these patient mutations and provide explanations as to why this part of the TRPML1 protein seemingly is such a hotspot for mutations affecting channel activity and how they explain this based on their structural evidence? What characteristics would be required for a small molecule agonist of TRPML1 in order to elicit larger activation in these patient LOF mutations if possible?

We thank the reviewer for highlighting these mutations identified in human patients. R403 appears to play two key roles. Firstly, its side chain participates in stabilizing Y404 in the open state. Secondly, as demonstrated in our previous study on TRPML1 (PMID: 35131932), the R403 side chain points towards the PI(3,5)P2 binding pocket, where it forms a critical salt bridge with the C3 phosphate group in the open state. Therefore, R403C mutation likely abolishes PI(3,5)P2 activation and also destabilizes the open state, resulting in the loss of function of the channel. We have expanded our discussion on this mutation in the revision. F408 is positioned at the junction between S4 and the S4-S5 linker. Its deletion mutation could change the stability or the folding of the protein. It is difficult to speculate the exact cause of the F408Δ LOF based on the TRPML1 structure. We don’t feel the effect of this mutation is relevant to the findings of this study.

(2) The authors used ML-SA1 only as a basis for their claims. Could they possibly provide some key data also on alternative small molecule agonists such as SF-51 and/or MK6-83?

We thank the reviewer for this suggestion. The TRPML1 agonists such as ML-SA1 (derived from SF-51) and MK6-83 have been well characterized in previous studies. In our study of sphingomyelin effect on TRPML1 activity, we used SF-51 to activate the channel (Figure 4b). The goal of our study is to demonstrate that agonist and antagonist can still allosterically regulate the LOF Y404A and GOF Y404W mutant channels, respectively, and their competition with sphingomyelin. We chose ML-SA1 in our experiment simply because it has been a commonly used TRPML1 agonist and its binding has been structurally defined, allowing us to compare various TRPML1 structures with different ligands. We don’t feel the use of other agonists would add extra information to our findings.

(3) Sphingomyelin effects on TRPML1 have been confirmed by other groups as well (see e.g. Prat Castro et al., 2022 https://pubmed.ncbi.nlm.nih.gov/36139381/) Fig.3. Interestingly TPC2 seems unaffected by sphingomyelin albeit it is also activated by PI(3,5)P2. Can the authors provide possibly some modeling and/or cryoEM data on TPC2 with sphingomyelin to potentially explain why TPC2 is seemingly unaffected by sphingomyelin?

We appreciate the reviewer for providing additional evidence of sphingomyelin's effects on TRPML1 and have included the reference in revision. The binding site and activation mechanism of PI(3,5)P2 are different between TPC2 and TRPML1. It is beyond the scope of this study and also too speculative to model sphingomyelin binding (if any) in TPC2 to explain its lack of effect on TPC2 activity.

**Reviewer #2 (Recommendations For The Authors):**
The findings from Gan et al provide structural insights into the allosteric regulation of TRPML1 channel gating. The authors have provided compelling and hard-won cryo-EM structural evidence of the channel regulation by PI4,5P2 and at sites that pack the gating pore of the VSDL (S4). However, as noted in the public review, the analysis of the cryo-EM structures that would support claims of open and closed channel states is woefully lacking. Additional information related to the functional results is required to evaluate the activation and inhibition kinetic effect of lipids and pharmacological agents used to support their allosteric mechanism of TRPML1 gating.Major concerns:(a) At the very least, the pore domains of the new channels (PDB 9CBZ and 9CC2) should be analyzed using the HOLE (or other) programs to estimate the distances along the ion-conducting pathway - are the structures wide enough to support the passage of hydrated or partially hydrated cations? Additional figure panels should provide this comparative analysis.

We thank the reviewer for this valuable suggestion. We have added a figure (Figure Supplement 3b) of pore domain radius analysis using the HOLE program in the revision and have also included the radius comparison with previous determined open and closed TRPML1 structures.

(b) At the very least, all current traces (Figures 1C, 1D, 1E, 4B, 4C) should be accompanied by time course plots of current amplitudes. It is impossible to evaluate the authors' claims of lipid and drug effects on TRPML1 channels without this information.

The corresponding time course plots of current amplitudes have now been included in the revision as Figure Supplement 1 and 7.

(c) Regarding the gain of function Y404W mutation structure, the authors' allosteric mechanistic hypothesis centers on side chain packing details within the VSD-like domain S4 (which in turn opens the pore through the S4-S5 linker). However, the local resolution within the structures at this site is not described. To assess the veracity of these claims, at the very least, authors should provide electron density maps of this region, either in Figure 2 or in Figure Supplement 4.

We have included the electron density map and local resolution information surrounding the W404 residue from the Y404W GOF mutant structure in the revision as Figure Supplement 3c.

Minor concerns:(d) Additional evidence related to the identity of the pore domain-associated lipid density (PC or sphingomyelin), and its channel regulation would improve the manuscript. The authors examine sphingomyelin, but what is the functional impact of PC on TRPML1 currents? While this is suggested, it is at the authors' discretion whether or not to carry out this analysis.

We thank the reviewer for raising this question. Adding extra PC has no effect on TRPML1 activity. This is expected since PC is the major lipid component of the membrane.

(e) The manuscript is well written. However, a few errors were noted while reviewing this draft.i. Line 138, (Figure F&G).ii. Line 66, "signaling transduction".

These errors have now been corrected.